

# Analysis of the complete genome sequence of a marine-derived strain *Streptomyces* sp. S063 CGMCC 14582 reveals its biosynthetic potential to produce novel anti-complement agents and peptides

Liang-Yu Chen[1,2], Hao-Tian Cui[1], Chun Su[3], Feng-Wu Bai[1] and Xin-Qing Zhao[1]

[1] State Key Laboratory of Microbial Metabolism and School of Life Science and Biotechnology, Shanghai Jiao Tong University, Shanghai, China
[2] School Life Science and Biotechnology, Dalian University of Technology, Dalian, Liaoning, China
[3] National Engineering Laboratory for Resource Developing of Endangered Chinese Crude Drugs in Northwest China, College of Life Sciences, Shaanxi Normal University, Xi'an, Shaanxi, China

## ABSTRACT

Genome sequences of marine streptomycetes are valuable for the discovery of useful enzymes and bioactive compounds by genome mining. However, publicly available complete genome sequences of marine streptomycetes are still limited. Here, we present the complete genome sequence of a marine streptomycete *Streptomyces* sp. S063 CGMCC 14582. Species delineation based on the pairwise digital DNA-DNA hybridization and genome comparison ANI (average nucleotide identity) value showed that *Streptomyces* sp. S063 CGMCC 14582 possesses a unique genome that is clearly different from all of the other available genomes. Bioactivity tests showed that *Streptomyces* sp. S063 CGMCC 14582 produces metabolites with anti-complement activities, which are useful for treatment of numerous diseases that arise from inappropriate activation of the human complement system. Analysis of the genome reveals no biosynthetic gene cluster (BGC) which shows even low similarity to that of the known anti-complement agents was detected in the genome, indicating that *Streptomyces* sp. S063 CGMCC 14582 may produce novel anti-complement agents of microbial origin. Four BGCs which are potentially involved in biosynthesis of non-ribosomal peptides were disrupted, but no decrease of anti-complement activities was observed, suggesting that these four BGCs are not involved in biosynthesis of the anti-complement agents. In addition, LC-MS/MS analysis and subsequent alignment through the Global Natural Products Social Molecular Networking (GNPS) platform led to the detection of novel peptides produced by the strain. *Streptomyces* sp. S063 CGMCC 14582 grows rapidly and is salt tolerant, which benefits efficient secondary metabolite production via seawater-based fermentation. Our results indicate that *Streptomyces* sp. S063 has great potential to produce novel bioactive compounds, and also is a good host for heterologous production of useful secondary metabolites for drug discovery.

Corresponding authors
Chun Su, suchun@snnu.edu.cn
Xin-Qing Zhao, xqzhao@sjtu.edu.cn

## INTRODUCTION

Marine streptomycetes are well-known producers of a myriad of useful secondary metabolites (*Subramani & Aalbersberg, 2012*). Genome sequences have been widely employed to unveil mechanisms of environmental adaption (*Ian et al., 2014*; *Tian et al., 2016*; *Undabarrena et al., 2017*), as well as to discover novel bioactive compounds by genome mining (*Gomez-Escribano, Alt & Bibb, 2016*; *Ma et al., 2017*; *Remali et al., 2017*). However, many available genome sequences of marine streptomycetes are only draft sequences (*Gomez-Escribano, Alt & Bibb, 2016*; *Ma et al., 2017*; *Remali et al., 2017*). The shortage of complete genome sequences negatively influences advanced genome-based studies. Currently, there are only seven publicly available genome sequences of marine *Streptomyces* (a total of 45 complete streptomycetes genomes and 260 scaffold streptomycetes genomes, data collected from the GenBank database on Aug 1, 2018). Given the importance of marine streptomycetes in the production of bioactive compounds, it is of great value to obtain more complete genome sequences.

Among various bioactivities of secondary metabolites from actinobacteria, anti-complement compounds are of great interest for drug discovery to treat numerous diseases resulting from inappropriate or excessive activation of the human complement system (*Morgan & Harris, 2015*). Until now, such compounds isolated from algae and plants have been studied (*Wen et al., 2017*; *Jin et al., 2015*), whereas compounds with such activities have been limitedly explored in microorganisms (*Xu et al., 2018*). So far, only two compound families from microorganisms have been reported to have anti-complement activity (ACA); one is complestatin and its analogs (*Kaneko, Kamoshida & Takahashi, 1989*), and the other is tunicamycin, which was reported in our recent work (*Kaneko, Kamoshida & Takahashi, 1989*).

Screening of microbial strains producing anti-complement compounds in our group revealed the potential of various marine-derived streptomycetes to produce anti-complement agents. Among these streptomycetes, *Streptomyces* sp. S063 distinguished from the other strains in its strong anti-complement activities. Here, we present the analysis of the complete genome sequence of this strain and analysis of its biosynthesis potential. In addition, four BGCs involved in biosynthesis of non-ribosomal peptides were disrupted, and anti-complement activities of the mutants were investigated. On the other hand, novel peptide was identified, and its biosynthetic genes was proposed. Our studies indicated that *Streptomyces* sp. S063 has the potential to produce novel anti-complement agents and peptides. The results in this work also imply that *Streptomyces* sp. S063 could be useful for production of various novel bioactive secondary metabolites.

## MATERIALS & METHODS

### Medium and microbial strain cultivation

*Streptomyces* sp. S063 was originally isolated from marine sediment collected in Xinghai Bay, Dalian, China. The strain was deposited in the China General Microbiological Culture Collection Center (CGMCC) with the accession number CGMCC 14582. The colony appearance was examined after incubation at 28 °C for 3 days on various agar media listed in Table S1.

   *Streptomyces* sp. S063 was inoculated into 20 mL of TSB medium in a 50 mL tube and incubated at 28 °C for 4 days to produce a seed culture. Carbohydrate utilization was tested on different solid media (TSB, A1, M2, M3, M8, M9, M12, M17, M19, M22 and cellulose-containing MS medium) (Table S1), culturing at 28 °C for 7 days. Tolerance to NaCl was tested on TSB agar plates containing 0, 3, 7, 9 and 11% (*w/v*) NaCl and cultured at 28 °C.

### Genome sequencing, assembly, annotation and mining

The 16S rRNA gene sequence of *Streptomyces* sp. S063 was obtained by PCR amplification using the methods described previously (*Li et al., 2007*) with the primers 27F (5′-AGAGTTTGATCCTGGCTCAG-3′) and 1429R (5′-AAGGAGGTGATCCAAGCCGCA-3′). The PCR product was sequenced, and the sequence was then uploaded to the web-based EzTaxon-e program (http://eztaxon-e.ezbiocloud.net/) (*Kim et al., 2012*) for further analysis. The phylogenetic trees based on the 16S rRNA gene sequences and 16S-23S rRNA internal transcribed spacer (ITS) sequences were created by the software Geneious (*Kearse et al., 2012*) based on the EzTaxon-e database and BLAST. The bootstrap values for phylogenetic analysis were based on 1,000 replicates.

   *Streptomyces* sp. S063 was grown in TSB medium at 28 °C for 3 days for genomic DNA extraction. High-quality genomic DNA was prepared manually using the methods described previously (*Lee et al., 2003*). The size, purity, and double-strand DNA concentration of the genomic DNA were measured by pulsed-field gel electrophoresis and the ratio of absorbance values at 260 nm and 280 nm, respectively, to assess the quality of genomic DNA. The genome was sequenced with the Pacbio technology (*English et al., 2012*) at Shanghai Jiao Tong University, yielding 1.25 Gb of raw data, which was assembled by Canu 1.4 (*Berlin et al., 2015*), yielding one single scaffold. The whole genome project has been deposited at GenBank under the accession number CP021707.

   The open reading frame (ORF) prediction and genome annotation were acquired by RAST (Rapid Annotation using Subsystem Technology) (*Aziz et al., 2008*; *Overbeek et al., 2013*; *Brettin et al., 2015*). Gene annotation was performed based on Clusters of Orthologous Groups (COGs), Kyoto Encyclopedia of Genes and Genomes (KEGG) and the Pfam database was performed through WebMGA (*Altschul et al., 1990*; *Ogata et al., 1999*; *Finn et al., 2016*). Identification of potential secondary metabolite biosynthetic gene clusters was achieved using antiSMASH (*Weber et al., 2015*), and confirmed with manual BLAST (Basic Local Alignment Search Tool) alignment.

   OrthoANI was used to generate the OAT heat map (*Lee et al., 2016*). ANI Calculator from EZBioCloud was used to compare the OrthoANIu value between *Streptomyces* sp. S063

and subspecies of *S. griseus* (*Yoon et al., 2017*). Digital DNA-DNA hybridization (dDDH) between the two genomes was determined by Genome-to-Genome Distance Calculator (GGDC) 2.1 (http://ggdc.dsmz.de/distcalc2.php) (*Auch, Klenk & Goker, 2010*). The selected genomes in the OAT heat map and the genomes of the four marine-derived streptomycetes with complete genomes in NCBI Genome database were chosen to compare with the genome of *Streptomyces* sp. S063 by BLAST Ring Image Generator (BRIG v0.95) with BLAST 2.5.0 (*Alikhan et al., 2011*).

## Construction of knockout mutants of *Streptomyces* sp. S063

Genetic manipulations to construct the knockout mutants were designed based on RecET direct cloning and conjugation protocols described elsewhere (*Wang et al., 2016a*; *Gust et al., 2003*). All primers used in this study are listed in Table S2. The apramycin resistance gene *aprA* was cloned with primers apra F/R from the plasmid pIJ773. The homologues DNA fragments, namely, 4-k1 and 4-k3 for *Cluster4*, 23-k1 and 23-k3 for *Cluster23*, 25-k1 and 25-k3 for *Cluster25* and 28-k1 and 28-k3 for *Cluster28* were obtained with the primers 4-k1 F/R, 4-k3 F/R, 23-k1 F/R, 23-k3 F/R, 25-k1 F/R, 25-k3 F/R, 28-k1 F/R and 28-k3 F/R, respectively. The *apra* resistance cassette with overlapping regions, namely, 4-apra, 23-apra, 25-apra and 28-apra, were acquired with the primers 4-773 F/R, 23-773 F/R, 25-773 F/R and 28-773 F/R. The pUC18 plasmid was digested with *Bam*HI, and the sequences with overlapping regions, namely, 4-18, 23-18, 25-18 and 28-18, were acquired with the primers 4-18 F/R, 23-18 F/R, 25-18 F/R and 28-18 F/R.

*Escherichia coli* GB05RedTrfA/pSC101-BAD-ETgA-tet was cultured in LB medium containing 4 $\mu$g/mL tetracycline at 30 °C overnight for seed culture. Then, 300 $\mu$L of the seed culture was inoculated into 15 mL LB medium containing 4 $\mu$g/mL tetracycline in a 100 mL flask and cultured at 30 °C and 300 rpm until the $OD_{600}$ value reached 0.35–0.4. Subsequently, 350 $\mu$L L-arabinose was added, and the broth was cultured at 37 °C for another 40 min. The *E. coli* GB05RedTrfA/pSC101-BAD-ETgA-tet were collected and re-dissolved into 200 $\mu$L ddH$_2$O. The collected four sequences for each cluster were mixed at a 2:2:2:5 (apra:1k:3k:18) ratio in an 11 $\mu$L volume and added into a 20 $\mu$L *E. coli* GB05RedTrfA/pSC101-BAD-ETgA-tet suspension for electroporation. The electroporated strains were spread on LB agar containing 100 $\mu$g/mL ampicillin and 50 $\mu$g/mL apramycin.

The constructed knockout plasmids, 4-pUC18, 23-pUC18, 25-pUC18 and 28-pUC18, were extracted and verified by DNA sequencing. The knockout plasmids were further electroporated into *E. coli* ET12567/pUZ8002 for further conjugation (*Gust et al., 2003*). The spores of *Streptomyces* sp. S063 were heated at 50 °C for 15 min and mixed with the *E. coli* strains and then spread on A1 media containing 10 mM MgSO$_4$. The mixed strains were cultured at 30 °C for 20 h, and the plates were then overlaid with 1 mL distilled water containing 0.5 mg nalidixic acid and 1.25 mg apramycin, after which the culture was continued for another 2–3 days. The selected mutants were verified with the primers 4yan F, 23yan F, 25yan F, 28yan F and 773yan R, and further confirmed by subsequent sequencing.

### ACA test of *Streptomyces* sp. S063

For the activity test, agar plates containing the culture of *Streptomyces* sp. S063 were extracted using 20 mL MeOH or 20 mL EtOAc, and then the resultant extracts were dried and re-dissolved into 1 mL MeOH or 1 mL water for use. The protocol to evaluate the ACA was based on the modified method from our laboratory (*Xu et al., 2018*). The absorbance at 405 nm of the supernatants (200 μL) was measured with a spectrophotometer (Multiskan GO 1510; Thermo Fisher Scientific, Vantaa, Finland).

### Detection of secondary metabolites produced by *Streptomyces* sp. S063

After culturing on A1 agar for 7 days, the agar containing the culture was cut into small pieces and extracted by MeOH: water (1:1), ACN: EtOAc (1:1) and EtOAc, respectively. The extracted samples were dried with a centrifugal dryer at 40 °C and dissolved into MeOH. The samples were injected into a Phenomenex Kinetex C18 (100Å, 1.7 μm, 50× 2.1 mm) column and analyzed using MS/MS Bruker microTOF-Q II (Bruker, Hamburg, Germany) coupled with HPLC (Agilent Infinity 1290) under the following LC analysis conditions: 0–1 min (5% ACN/$H_2O$ with 0.1% formic acid), 1–9 min (a gradient of ACN/$H_2O$ with 0.1% formic acid from 5% to 100%), 9–10 min (100% ACN with 0.1% formic acid) with a 0.5 mL min$^{-1}$ flow rate. The microTOF-Q II setting during the LC gradient was as follows: positive ion mode mass range 200–2,000 m/z, MS scan rate 1/s, MS/MS scan rate 10/s. The acquired mass spectrometry data was uploaded and compared with GNPS (Global Natural Products Social Molecular Networking) (*Wang et al., 2016b*). The generated data from GNPS on secondary metabolites was visualized by Cytoscape v3.4.

## RESULTS

### Characterization of the growth properties and salt tolerance of *Streptomyces* sp. S063

*Streptomyces* sp. S063 grows well at 28 °C on culture media containing TSB, A1, M2, M3, M8, M17, M19 and MS with cellulose, but grows poorly on M12 (Fig. 1A). No growth was detected on M9 and M22 medium. White gray spores can be observed within less than 3 days after cultivation in a proper medium (A1 or MS), which normally takes 7 days for the common streptomycetes, indicating its fast growth rate. The aerial mycelia are branched and yellowish white in color on A1, TSB or MS and become orange or dark red on M2, M3 or M19 (Fig. 1). Salinity tolerance test indicates that *Streptomyces* sp. S063 survives in TSB medium containing up to 9.5% NaCl (Table S3), and the strain showed optimum growth in the presence of 3% NaCl.

### 16S rRNA gene sequence analysis of *Streptomyces* sp. S063

Phylogenetic analysis based on the 16S rRNA gene sequences revealed that *Streptomyces* sp. S063 may be a subspecies of *S. badius* NRRL B-2567[T] (100% similarity) (*Pridham, Hesseltine & Benedict, 1958*) (Fig. 2), but so far, no genome sequences of *S. badius* NRRL B-2567[T] are available. The genome sequence of *Streptomyces* sp. S063 thus provides valuable information for genome-based studies of *S. badius* and its phylogenetically related strains.

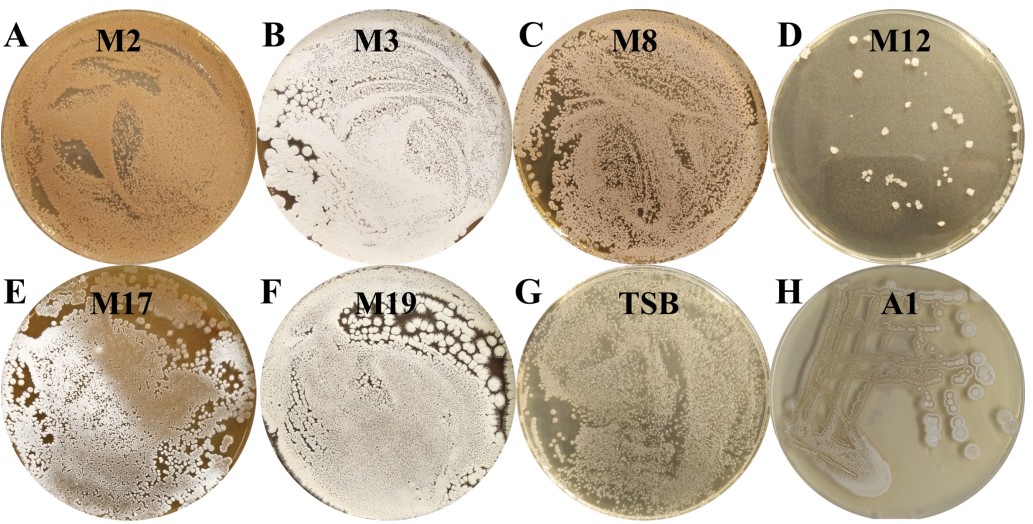

**Figure 1** The pictures of *Streptomyces* sp. S063. (A–H): agar media M2, M3, M8, M12, M17, M19, TSB and A1. *Streptomyces* sp. S063 was cultured at 28 °C for 3 days, and the plates were photographed.

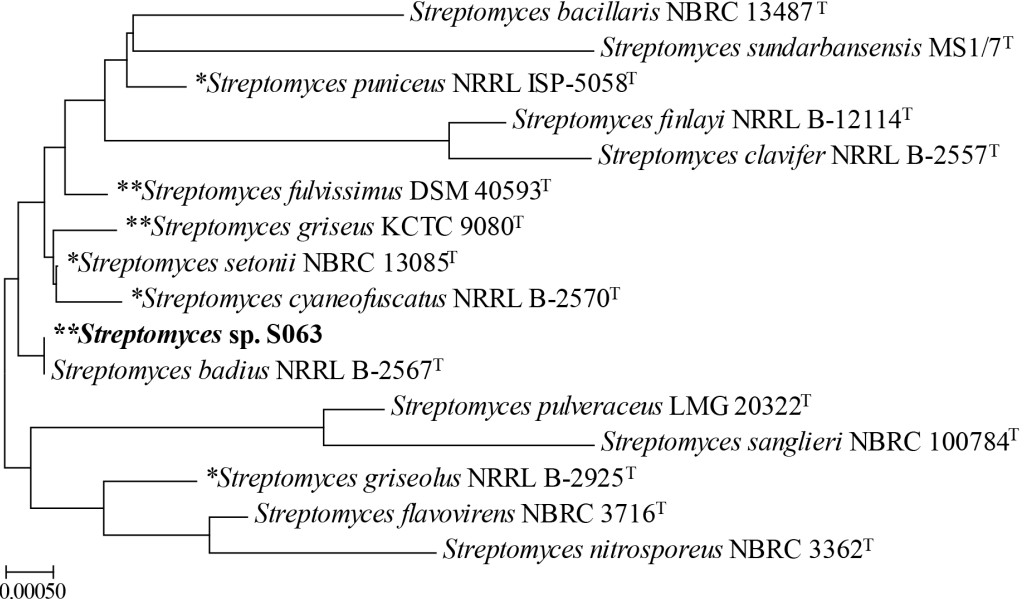

**Figure 2** Neighbor-joining tree showing the phylogenetic relationships of *Streptomyces* sp. S063 with the related reference strains based on 16S rRNA gene sequences. The strains were selected based on the alignment in the EzBiocloud 16S database and the NJTree was constructed by Geneious. *, strain with contig type genome; **, strain with complete genome.

Besides, a Neighbor-joining (NJ) Tree based on the 16S-23S rRNA ITS sequences was constructed for phylogenetic analysis of *Streptomyces* sp. S063 (Fig. S2). However, due to the limited availability of ITS sequences, the ITS NJTree did not tell more specific relationships with other strains than the 16S rRNA NJTree.

**Table 1 General genome features and genome sequencing information of *Streptomyces* sp. S063.**

| Features | Chromosome |
|---|---|
| Genomic size (bp) | 7,614,683 |
| GC content | 71.49% |
| ORFs | 7,109 |
| rRNA | 18 |
| tRNA | 65 |
| NCBI project ID | PRJNA388536 |
| Accession number | CP021707 |
| Sequencing platforms | PacBio RS |
| Fold coverage | 150× |
| Assemblers | Canu v1.4 |
| Annotation source | RAST |

## General genome information

The yielded 1.25 Gb raw data was assembled to generate the 7,614,683 bp genome of *Streptomyces* sp. S063 (Table 1). The genome includes 7,109 protein-encoding genes, 18 rRNA genes and 65 tRNA genes covering 86.3% of the whole genome. No plasmid sequence was detected in the sequence data. The GC content of the whole genome is 71.85% (Table 1). In total, 4,369 putative genes were classified into functional categories based on clusters of orthologous genes (COG) designation (Table S4). The most abundant categories are transcription (683), carbohydrate transport and metabolism (456) and amino acid transport and metabolism (462). There are an additional 753 genes that belong to the category of general function prediction only. On the other hand, there are 423 genes with yet unknown functions. The circular representation of the genome and the distribution of the COG categories are shown in Fig. 3.

## Genome sequence comparison analysis

Further comparison of the genome of *Streptomyces* sp. S063 was performed to evaluate how different this genome is from the available genomes. From the OAT heat map genome comparison (*Lee et al., 2016*), *Streptomyces* sp. S063 has the highest ANI (average nucleotide identity) value of 91.5% compared with *S. setonii* NRRL ISP-5322 [T] (contig type genome), and the second highest to *S. griseus* NBRC 13350 (complete genome, ANI value 90.4%, Fig. 4). The results of the OrthoANIu value between *Streptomyces* sp. S063 and all of the subspecies of *S. griseus* NBRC 13350 are all below the proposed species boundary cut-off value of 95–96% (Table S5) (*Kim et al., 2014*). We further compared the complete genomes of *Streptomyces* sp. S063 and *S. griseus* NBRC 13350. The calculated DDH estimates were based on high-scoring segment pairs (HSPs), including HSP length/total length (53.50%), identities/HSP length (39.90%), and identities/total length (50.50%) and are all far below 70%, which is the cut-off value for species delineation. The whole genome comparison between *Streptomyces* sp. S063 and other strains is presented in Fig. S1.
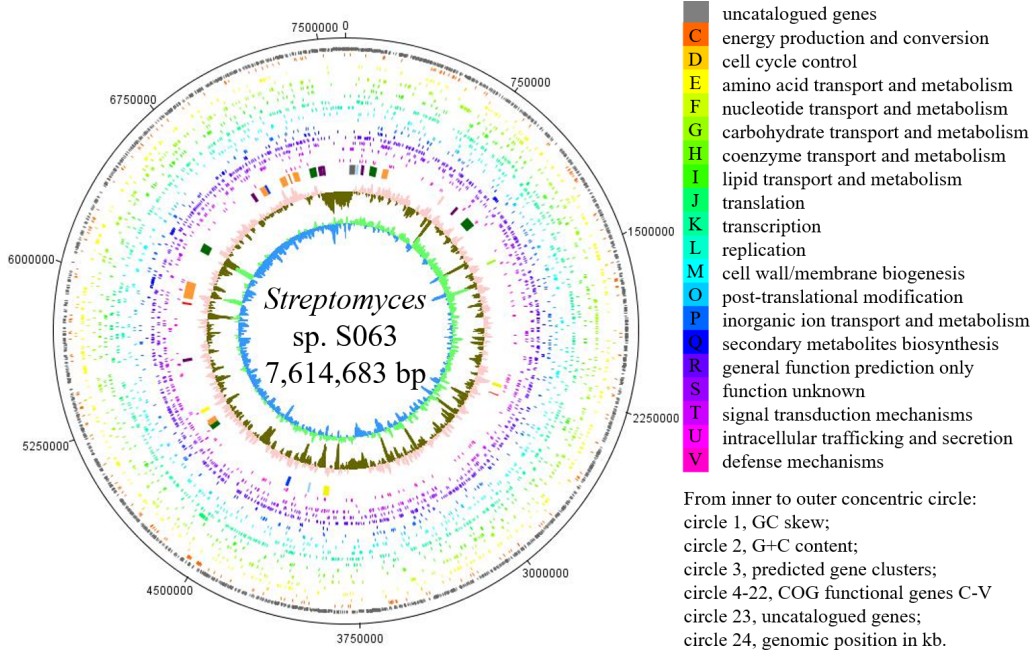

**From inner to outer concentric circle:**
circle 1, GC skew;
circle 2, G+C content;
circle 3, predicted gene clusters;
circle 4-22, COG functional genes C-V
circle 23, uncatalogued genes;
circle 24, genomic position in kb.

**Figure 3** **Circular representation of the *Streptomyces* sp. S063 genome.** Labeling from the inside to outside circle: circle 1, GC skew (calculated as $(G + C)/(G - C)$, green/blue peaks out/inside the circle indicates values higher or lower than 1, respectively); circle 2, G + C content (peaks out/inside the circle indicate values higher or lower than average G + C content, respectively); circle 3, predicted BGCs; circle 4-22, COG functional genes C-V; circle 23, uncharacterized genes; circle 24, genomic position in kb.

## Biosynthetic gene clusters in the genome of *Streptomyces* sp. S063

There are 243 genes whose function was assigned in the category of secondary metabolites biosynthesis, transport and catabolism, comprising 3.24% of the *Streptomyces* sp. S063 genome. Genome sequence analysis of *Streptomyces* sp. S063 revealed the presence of at least 30 candidate biosynthetic gene clusters (BGCs). The BGCs that are potentially involved in the biosynthesis of secondary metabolites are presented in Table 2. The distribution of the predicted BGCs was listed in Fig. 3. These 30 BGCs comprise 12 different types of BGCs, including BGCs containing non-ribosomal peptide synthetase (NRPS), polyketide synthetase (PKS), lanthipeptide, terpenes, lassopeptide, butyrolactone, nucleoside, bacteriocin, ectoine, melanine, siderophores and hybrid BGCs. There are five BGCs showing no similarity to any reference BGCs, and 13 BGCs containing over 80% genes showing high similarities to the reported BGCs.

There are three BGCs which are involved in the biosynthesis of lantipeptides in the genome. *Cluster10* and *Cluster12* are related to novel type II lantipeptides as indicated by the sequence analysis. Through comparison, *Cluster10* may be responsible for the synthesis of two candidate products, whereas *Cluster12* is probably responsible for the four proposed compounds. *Cluster17* is highly similar to that of the class III lantipeptide of AmfS (Table S6), which belongs to a biological surfactant that positively regulates the formation of aerial mycelia (*Ueda et al., 2002*).

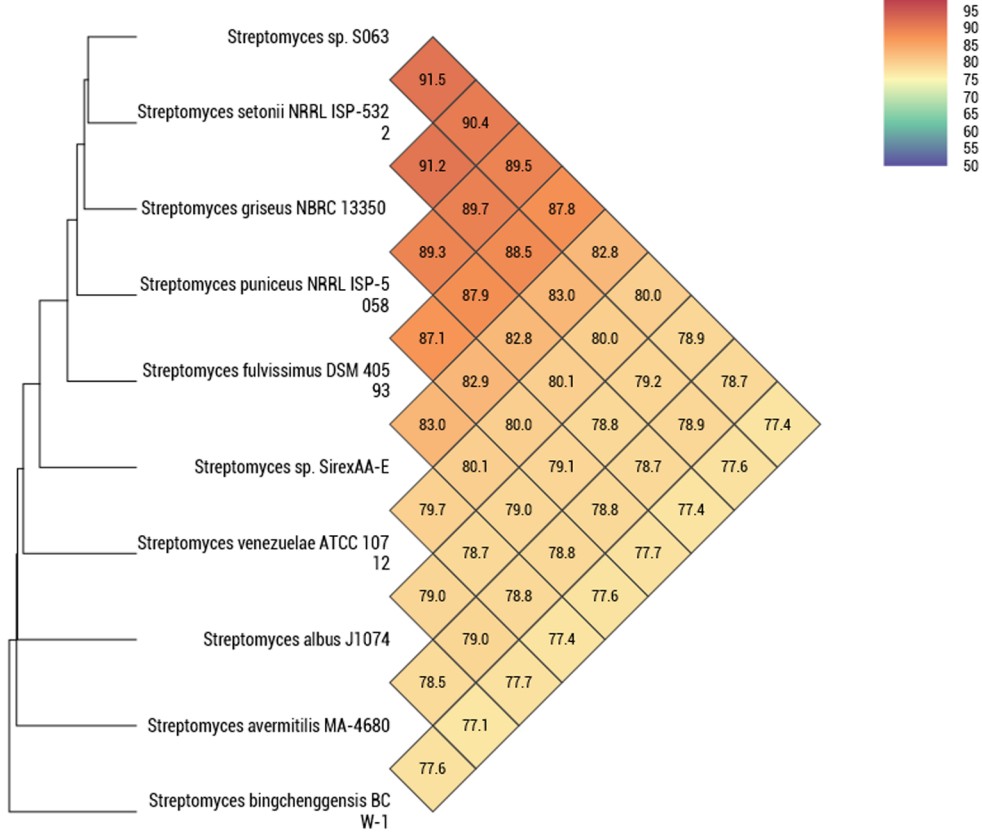

**Figure 4  OATtree of *Streptomyces* sp. S063 based on comparison of the complete genome sequences.**
All of the other nine genome sequences were acquired from the GenBank and were compared to the
genome of *Streptomyces* sp. S063 with both directions through OrthoANI tool version 0.93.

Among the five NRPS-related BGCs, *Cluster4*, *Cluster8* and *Cluster23* have relatively high
similarities to the reference BGCs. *Cluster4* is close to the peptide siderophore coelichelin
BGC with genes sharing 77–87% similarities (*Corre & Challis, 2009*). *Cluster8* is involved
in the biosynthesis of a non-ribosomal peptide that is similar to stenothricin (*Liu et al.,
2014*), an inhibitor of bacterial cell wall biosynthesis (Table S7). As most genes in these two
BGCs share more than 90% similarity (*Liu et al., 2014*), and the biggest difference is only
the first amino acid (Cys *vs.* Asn), we deduced that the final structures should be similar.
*Cluster23* is a PKS-NRPS hybrid BGC and is assumed to be close to SGR_PTMs (*Luo et al.,
2013*).

Cluster21 is designated as Bacteriocin-NRPS hybrid BGC, which might be responsible for
a ten amino acid cyclic peptide. In the following GNPS analysis, the discovery of surfactin
analogs seems to be associated with this BGC. However, this BGC has a low similarity to
all of the reported BGCs including the surfactin BGC. The majority of biosynthetic genes

**Table 2   Potential BGCs in the genome of *Streptomyces* sp. S063.**

| No. | Type | Length (bp) | Similar BGC (gene number) | [a]Genes with similarity (%) | Similarities (%) |
|---|---|---|---|---|---|
| 1 | Other | 42,678 | Nocathiacin | 4 | |
| 2 | Butyrolactone | 2,217 | Butyrolactone (2) | 100 | 66–73 |
| 3 | Terpene | 22,022 | – | | |
| 4 | NRPS | 23,005 | Coelichelin (11) | 81 | 77–87 |
| 5 | T3PKS | 5,025 | Naringenin (2) | 100 | 58–71 |
| 6 | Nucleoside | 20,719 | Toyocamycin (10) | 20 | 60–68 |
| 7 | Terpene | 21,213 | Steffimycin (37) | 19 | 77–92 |
| 8 | NRPS | 75,710 | Stenothricin (22) | 90 | 90–98 |
| 9 | Ectoine | 3,702 | Ectoine (4) | 100 | 94–99 |
| 10 | Lantipeptide | 23,227 | – | | |
| 11 | Siderophore | 11,778 | Desferrioxamine (5) | 80 | 92–95 |
| 12 | Lantipeptide | 39,575 | Chalcomycin | 9 | |
| 13 | Butyrolactone-Ectoine | 15,263 | Pristinamycin | 17 | 46–61 |
| 14 | Lassopeptide | 4,996 | SRO15-2005 (5) | 100 | 93–98 |
| 15 | NRPS-T1PKS | 54,557 | WS9326 | 7 | |
| 16 | T2PKS | 27,055 | Alnumycin (32) | 84 | 54–95 |
| 17 | Lantipeptide | 7,779 | AmfS (5) | 100 | 93–100 |
| 18 | Terpene | 21,190 | Laspartomycin | 9 | |
| 19 | Siderophore | 14,729 | – | | |
| 20 | T1PKS-Oligosaccharide | 124,306 | Incednine (73) | 41 | 42–81 |
| 21 | Bacteriocin-NRPS | 64,569 | Enduracidin | 8 | |
| 22 | Terpene | 26,570 | Hopene (13) | 69 | 69–93 |
| 23 | T1PKS-NRPS | 17,419 | SGR_PTMs (7) | 100 | 83–95 |
| 24 | Bacteriocin | 11,910 | – | | |
| 25 | T1PKS-NRPS | 50,642 | ECO-02301 | 7 | |
| 26 | Melanin | 1,677 | Melanin (2) | 100 | 77–91 |
| 27 | T3PKS | 4,701 | Alkylresorcinol (3) | 100 | 88–94 |
| 28 | NRPS | 52,238 | Daptomycin | 7 | |
| 29 | Terpene | 26,745 | Isorenieratene (7) | 100 | 77–90 |
| 30 | Terpene | 22,084 | – | | |

**Notes.**

[a]The percentage of genes in reference BGC showing similarity to the subject BGC, which is calculated by antiSMASH.

are with identifies around 50%. *Cluster21* is consisted of 12 continuous NRPS genes with 10 possible A-domains, which is absent in the similar region of the closely-related genome of *S. griseus* NBRC 13350[T]. There are four A-domain genes in *Cluster21* that are predicted to be responsible for the Leu residue, which partially match with the peptide sequence of the surfactin analogs. Intriguingly, the peptide analogs we detected are composed of seven amino acid residues, but there are 10 predicted A-domains in the *Cluster21* BGC.

In the seven PKS-associated BGCs, *Cluster5* and *Cluster27* are type 3 and similar to BGCs naringenin and alkylresorcinol BGCs. *Cluster20* is similar to that of incednine (Table S8), which is a potential antitumor compound (*Takaishi, Kudo & Eguchi, 2013*). Although the two BGCs have quite different gene arrangements, most of the functional genes in *Cluster20*

are similar to that in the incednine BGC, and therefore similar products may be produced. However, comparing with that of the incednine BGC, ten functional genes are missing in *Cluster20*, including *IdnM1~5*, *IdnS5*, *IdnS12* and *Idne*. The loss of *IdnM1~5* responsible for methoxymalonyl-ACP biosynthesis could result in the absence of methoxy during the synthesis in the PKS module 10 of *IndP5*. *IndS5* encodes a potential methyltransferase, and the absence of this gene may lead to failure of transfer UDP-xylosamine to UDP-N-methylxylosamine. On the other hand, the absence of NDP-hexose-3,4-dehydratase *IndS12* would lead to one more hydroxyl group on the intermediate product TDP-N-demethylforosamine. *IndS15* encodes a potential glycosyltransferase and may have a similar function to *IndS14*, and the absence of this gene might affect the attachment of a glycosyl group. *Cluster16* is similar to that of alnumycin, which can inhibit the growth of *Bacillus subtilis* (Oja et al., 2008). The differences between these two BGCs are the absence of two components of the sensory kinase *gene* R4 and a different oxidase encoding gene *gene* 7 (31% identity), but it is still not clear what roles these two genes play in metabolites production.

The rest of the BGCs include 6 terpene, 1 lassopeptide, 2 bacteriocin and 2 siderophore BGCs. The product of *Cluster14* may be quite close to SRO15-2005, which belongs to class II lassopeptides acting as an antimicrobial, prolyl endopeptidase inhibitor and endothelin type-B receptor antagonist (Maksimov, Pan & James, 2012). *Cluster22* and *Cluster29* are terpene BGCs and are assumed to be close to that of hopene (Bentley et al., 2002) and isorenieratene BGCs (Krugel et al., 1999).

## Genome features related to marine adaptation

The marine adaptation transporters (MATs) (Tian et al., 2016) were analyzed to evaluate the marine adaption ability of *Streptomyces* sp. S063. There are 12 BCCT (betaine/carnitine/choline transporter) genes and 10 NhaA ($Na^+$: $H^+$ antiporters) genes in the genome of *Streptomyces* sp. S063. Moreover, the other MATs genes in *Streptomyces* sp. S063, including Trk ($K^+$ transporter), Tat (twin arginine targeting) and MOP (multidrug/oligosaccharidyl-lipid/polysaccharides) genes, are also more abundant than those in the two closest strains and are similar to that in the strains isolated from the South China Sea (Tian et al., 2016). Other transporter genes encoding RhtB (resistance to homoserine/threonine), ThrE (threonine/serine exporter), MscL (large conductance mechanosensitive ion channel), GPTS (general phosphotransferase system), ACR3 (arsenical resistance-3), MscS (small conductance mechanosensitive ion channel) and GntP (gluconate: $H^+$ symporter), which are absent in the two closest strains, are present in *Streptomyces* sp. S063. It was found that more such transporters are present in the genome of *Streptomyces* sp. S063 than its closest terrestrial counterparts, namely, *S. griseus* and *S. fulvissimus* (Table S9). Genes encoding the multicomponent $Na^+$:$H^+$ antiporter subunit proteins A~G, which are responsible for pumping out the surplus $Na^+$ in *Streptomyces* sp. S063, may contribute to the high salt tolerance of *Streptomyces* sp. S063 (Tian et al., 2016). The presence of the BGCs encoding the subunits A–N of NADH-quinone oxidoreductase (gene-4301~4314), which are typical for marine microorganisms, also agree with the marine origin of *Streptomyces* sp. S063 (Kube et al., 2013).

## ACA of *Streptomyces* sp. S063

We tested various media listed in Table S1 to evaluate their ACA. It was found that the MeOH extracts of the solid culture of the strain showed good ACA, while there was no ACA in the EtOAc extract samples. This indicated that the active agents would be polar which are easy to be acquired by polar solvents. We found that samples collected from media M2, M8, M17and M19 showed good ACAs, which were 59.3%, 95.3%, 74.4% and 48.2%, respectively, and the culture collected from medium M8 produced the best results (95.3%) (Fig. 5A). In contrast, although the strain grows well on M3 medium, no ACA was detected from the sample collected from this medium.

Because it has been found that the anti-complement agents from *Streptomyces* sp. S063 should be polar in this study and the complestatin analogs are NRP type compounds, we randomly selected five NRPS related BGCs, namely, *Cluster4*, *Cluster15*, *Cluster24*, *Cluster25* and *Cluster28* to identify the possible BGC responsible for the ACA. These five NRPS related BGCs were supposed to synthesize small peptide compounds which were likely to be polar. We successfully knocked out four BGCs, namely, *Cluster4*, *Cluster24*, *Cluster25* and *Cluster28*, and the acquired mutants were verified (Figs. 5B–5E). However, we did not observe any decrease in ACA in the four knockout mutants (∆*Cluster4*, 93.78, 91.55; ∆*Cluster24*, 91.62, 96.36; ∆*Cluster25*, 91.20, 93.04; ∆*Cluster28*, 92.01, 95.58) when compared to that of the wild-type strain, the ACAs of the wild-type strain and the four mutants in the M8 medium were all similar (91–97%). These results suggest that these BGCs is not responsible for the ACA, and the BGC responsible for the biosynthesis of novel anti-complementary agents remain to be explored.

## Identification of novel peptides produced by *Streptomyces* sp. S063

The GNPS map revealed that there are three large clusters and several small clusters including 75 nodes (Fig. S4). The biggest cluster containing 10 nodes and the third biggest cluster containing 7 nodes were predicted to be peptide type metabolites, but the predicted peptide sequences based on the MS/MS spectra have not been aligned with any known compounds or the BGCs in the genome. The second biggest cluster of metabolites showed no relationship with the peptide compounds. Among the rest of the MS/MS spectra, it was found that one cluster of spectra containing three nodes in the box in Fig. S3, namely, *m/z* 1,022 (one node) and 1,036 (two nodes with the same *m/z*), are quite similar to those of pumilacidins and surfactins. In comparison, the detailed MS/MS peaks revealed that the peptide sequence of *m/z* 1,022 detected in *Streptomyces* sp. S063 should be E-L-L-D-V-L-I (Fig. 6), and it is different from that of pumilacidin A (E-L-L-L-D-L-I, *m/z* 1,050) and surfactin C14 (E-L-L-V-D-L-I, *m/z* 1,036). It was deduced that the difference between the two compounds from *Streptomyces* sp. S063 was that the E residue in the peptide *m/z* 1022 turns into E+ *m/z* 14 in the compound *m/z* 1,036, which might be the mass of methoxy-Glu. Based on the comparisons when searching the SciFinder database, it was proposed that the two peptides might be novel and deserve further investigation. However, the analysis of the BGCs did not show obvious surfactin-like BGC. It was found that the BGC of *Cluster21*, which was predicted as a bacteriocin-NRPS BGC, could possibly produce the surfactin analogs.

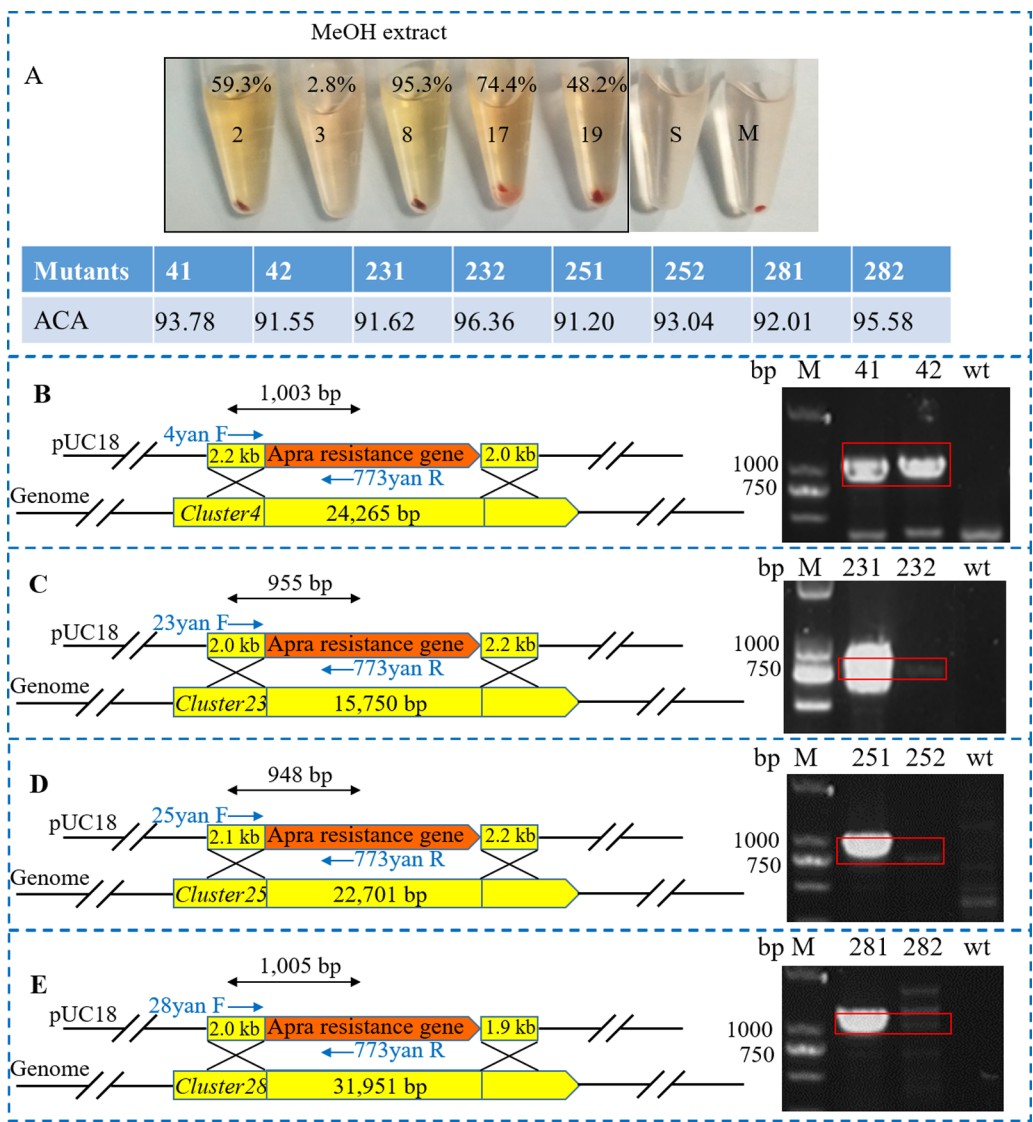

**Figure 5  Anti-complement activity and the knockout mutant construction of *Streptomyces* sp. S063.**
(A) anti-complement activities of *Streptomyces* sp. S063. MeOH extracts of the culture from the M2, M3, M8, M17 and M19 medium were tested, and the ACA values are presented. S: Standard hemolysis; M: Mechanical hemolysis. (B–E) The scheme of knockout of *Cluster4*, *Cluster23*, *Cluster25* and *Cluster28* and the corresponding PCR verification of the mutants.

## DISCUSSION

So far the complete genome sequences of marine streptomycetes have been still poorly available. Here, we present the complete genome sequence of a marine-derived streptomycete *Streptomyces* sp. S063 and analysis of the genome sequence. The whole genome comparison with other strains (Fig. S1) reveals that there are many small different sections distributed all over the genome. In addition, the annotations by COG and KEGG database indicated that it contains nearly one third unknown genes that require functional

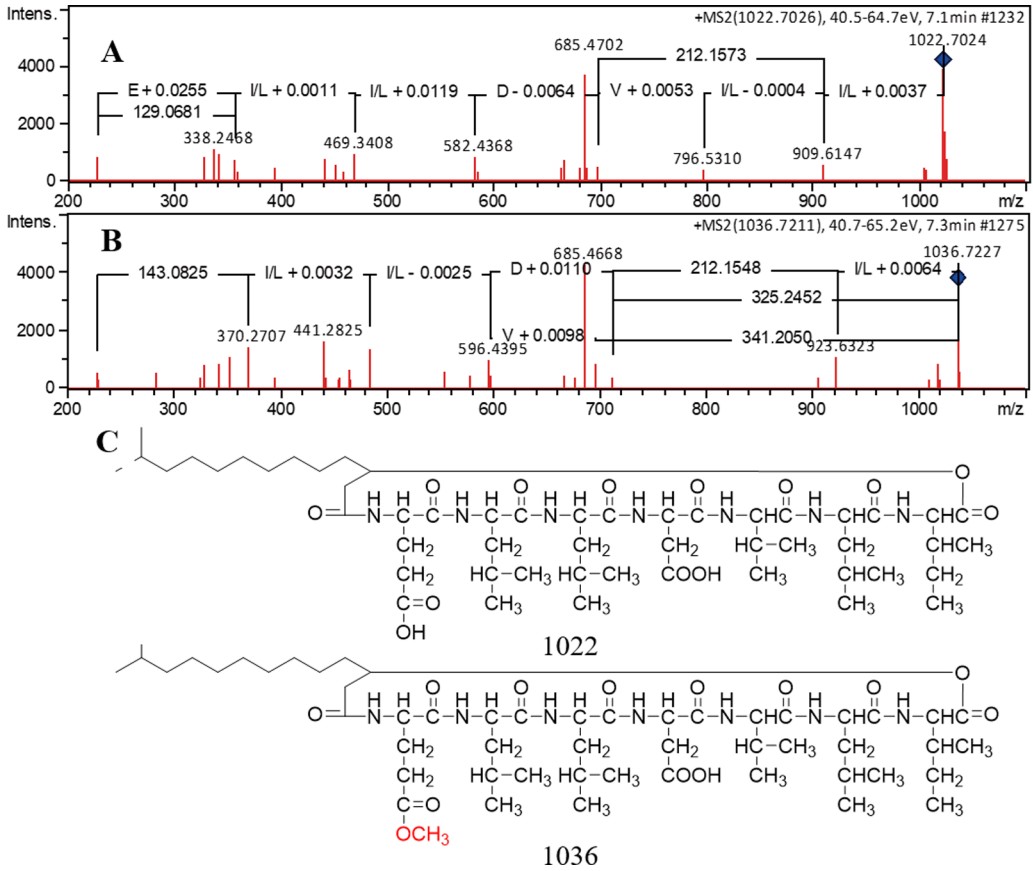

**Figure 6 The MS/MS spectra of predicted pumilacidin analogs and the predicted structures.** (A) The MS/MS spectrum of compound *m/z* 1,022; (B) The MS/MS spectrum of compound *m/z* 1,036; (C) The predicted structures of the two novel peptides. The extracted samples were dried with a centrifugal dryer at 40 °C and dissolved into MeOH. The samples were injected into a Phenomenex Kinetex C18 (100 Å, 1.7 μm, 50 × 2.1 mm) column and analyzed using MS/MS Bruker microTOF-Q II (Bruker, Hamburg, Germany) coupled with HPLC (Agilent Infinity 1290) under the following LC analysis conditions: 0–1 min (5% ACN/H$_2$O with 0.1% formic acid), 1–9 min (a gradient of ACN/H$_2$O with 0.1% formic acid from 5% to 100%), 9–10 min (100% ACN with 0.1% formic acid) with a 0.5 mL min$^{-1}$ flow rate. The microTOF-Q II setting during the LC gradient was as follows: positive ion mode mass range 200–2,000 *m/z*, MS scan rate 1/s, MS/MS scan rate 10/s.

analysis. Therefore, *Streptomyces* sp. S063 has a unique genome that could provide novel and valuable insights for various genome-based studies.

*Streptomyces* sp. S063 showed strong anti-complement activities, and therefore we focused on the BGC encoding the anti-complement agents. The majority of the current clinical anti-complement drugs are chemically synthesized, which have various side effects. The natural source-derived anti-complement drugs are mainly from plants (*Xu, Chen & Zhao, 2015*). So far, the known BGCs for microbial-derived anti-complement agents include the complestatin BGC (*Chiu et al., 2001*) and tunicamycin BGC (*Xu et al., 2018*). The complestatin BGC is an NRPS BGC and the tunicamycin BGC is a nucleoside BGC. However, no BGC showing any similarity to that of complestatin or tunicamycin was
identified in the genome of *Streptomyces* sp. S063, suggesting that the strain can produce previously unknown microbial anti-complement agents, which is the target of our future work. From our experience on manipulating various streptomycetes (*Chen et al., 2018*; *Kong et al., 2013*; *Su et al., 2015*), we found that *Streptomyces* sp. S063 is relatively easy to be genetically manipulated, and it will be promising not only for investigation of the BGC(s) involved in the production of anti-complement agents, but also for acting as a host strain for the production of novel secondary metabolites.

During the optimization of the culture media, we observed that *Streptomyces* sp. S063 grew rapidly in the media containing starch. Within 3 days, the strain could spread all over the plates and generate abundant spores, thus the cultivation of *Streptomyces* sp. S063 in M3, M19 and MS media containing starch is recommended for spore harvest and further conjugation experiments. The rapid growth and abundant spores facilitate easy genetic manipulation of *Streptomyces* sp. S063.

The GNPS platform has been proven to be useful for the analysis of natural products (*Crusemann et al., 2017*; *Mohimani et al., 2017*; *Wang et al., 2016b*). GNPS has been successfully applied in the identification of non-ribosomal peptides, lipopeptides and polyketides, such as retimycin A (*Duncan et al., 2015*), alterochromide (*Ross et al., 2015*), and columbamides (*Kleigrewe et al., 2015*). Our results employing GNPS and molecular networking (Fig. S4) indicate that *Streptomyces* sp. S063 may produce special secondary metabolites, which warrant further investigation. There are five clusters containing 23 nodes with molecular weight more than *m/z* 1,000, which might be associated with peptide type metabolites and the rest detected nodes are mostly larger than *m/z* 500. The number and average molecular weight of the detected nodes are larger and higher than most of other strains that we have investigated, which would contribute more complex metabolites for research. However, there are still great challenges to match the MS/MS spectra with the genome information. The annotated and public MS/MS spectrum resources, including the GNPS database, are still limited compared with the vast number of natural product families, which restricted the annotation of the molecular networking clusters. In addition, even though the BGC prediction bioinformatics software has been rapidly developed, it is still difficult to make an exact prediction of gene function and putative products, especially when facing novel biosynthetic pathways. On the other hand, the limitation of the MS/MS spectra in elucidation of specific structures, especially to distinguish them from different analogs, also demands the assistance of other structural elucidation methods. It is expected that the discovery of novel bioactive metabolites can be facilitated by further development of advanced analysis methods and bioinformatics tools, as well as enrichment of the public database.

The surfactin or pumilacidin analogs are reported as antibacterial, antifungal and antivirus compounds produced by *Bacillus* (*Roongsawang, Washio & Morikawa, 2010*). In this study, it is interesting to observe some analogs of surfactin in the A1 agar medium of *Streptomyces* sp. S063 through the molecular networking analysis (Fig. S4). The observed analogs were proposed to be novel and were proposed to be associated with the novel BGC *Cluster21*. Although we cannot fully explain how the ten predicted A-domains yield seven amino acids in the surfactin analogs, we found reports in the literature indicating that not

all of the A-domains are involved in the biosynthetic pathway in the surfactin analog BGCs (*Saggese et al., 2018*). Thus, it would be valuable to study the products of *Cluster21* to verify their relationship to surfactins, which might involve novel biosynthetic mechanisms.

The salinity tolerance test indicates that *Streptomyces* sp. S063 could survive in the media with up to 9.5% NaCl, and the rapid growth of *Streptomyces* sp. S063 in the presence of NaCl suggests that this strain can utilize seawater for bioproduction. It is widely acknowledged that the drinkable water is becoming limited and submerged fermentation would consume large quantity of water. Therefore, fermentation using salt-tolerant microbial strains is of great interest for modern industrial biotechnology (*Chen & Jiang, 2018*). *Streptomyces* sp. S063 is promising to be further developed as a cell factory that can be operate in seawater for bioactive compounds production.

## CONCLUSIONS

In summary, the 7.6 Mb complete genome of *Streptomyces* sp. S063, which produces anti-complement agents, provides valuable data for genome-based studies of marine streptomycetes. Analysis of the genome identified at least 30 secondary metabolite BGCs, and the presence of yet-unknown BGCs that are responsible for the production of previously unknown anti-complement agents of microbial origin was confirmed. *Streptomyces* sp. S063 has the potential to be developed as a producer for novel secondary metabolites production due to the presence of novel BGCs and high efficiency of genetic manipulation. The salt tolerance property of this strain can also be utilized to produce useful bioproducts using seawater.

## ACKNOWLEDGEMENTS

We appreciate Prof. Pieter C. Dorrestein from the University of California, San Diego, USA, for the kind help with the MS/MS data acquisition.

### Funding
This work was supported by the Fundamental Research Funds for the Central Universities (GK201703035), the State Key Laboratory of Tropical Marine Biology Resources and Ecology, Chinese Academy of Sciences (LMB111002), the Key Research Development Program of the Natural Science Basic Research Plan in Shaanxi Province of China (2018NY-106) and the National Natural Science Foundation of China (No. 31600038). The funders had no role in study design, data collection and analysis, decision to publish, or preparation of the manuscript.

### Grant Disclosures
The following grant information was disclosed by the authors:
Fundamental Research Funds for the Central Universities: GK201703035.

State Key Laboratory of Tropical Marine Biology Resources and Ecology, Chinese Academy of Sciences: LMB111002.

Key Research Development Program of the Natural Science Basic Research Plan in Shaanxi Province of China: 2018NY-106.

National Natural Science Foundation of China: 31600038.

## Competing Interests

The authors declare there are no competing interests.

## Author Contributions

- Liang-Yu Chen conceived and designed the experiments, performed the experiments, analyzed the data, contributed reagents/materials/analysis tools, prepared figures and/or tables, authored or reviewed drafts of the paper, approved the final draft.
- Hao-Tian Cui performed the experiments, approved the final draft.
- Chun Su prepared figures and/or tables, authored or reviewed drafts of the paper, approved the final draft.
- Feng-Wu Bai supervised the research and revised the manuscript the drafts of the manuscript, approved the final draft.
- Xin-Qing Zhao conceived and designed the experiments, prepared figures and/or tables, supervised the research and revised the manuscript, approved the final draft.

## DNA Deposition

The following information was supplied regarding the deposition of DNA sequences:

NCBI GenBank accession: PRJNA388536.

The GNPS data is accessible here: https://gnps.ucsd.edu/ProteoSAFe/result.jsp?task=17a83d1366f849a29e063bd1d3e81c36&view=advanced_view.

## Data Availability

The raw data is included in Figs. 1–6 and in the Supplemental File.

## Supplemental Information

Supplemental information for this article can be found online at http://dx.doi.org/10.7717/peerj.6122#supplemental-information.

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
