# Peer review of "Analysis of the complete genome sequence of a marine-derived strain Streptomyces sp. S063 CGMCC 14582 reveals its biosynthetic potential to produce novel anti-complement agents and peptides"

_PeerJ, doi:10.7717/peerj.6122_

## Round 0.1 · original submission · Major Revisions

Both reviewers have raised important concerns that should be addressed before proceeding with the peer review process.

Reviewer 1 ·

Basic reporting

No comment

Experimental design

No comment

Validity of the findings

No comment

Additional comments

The manuscript by Chen et al. describes the genome analysis and biosynthetic potential of a marine Streptomyces strain. The genome of this new strain was sequenced and analysed with bioinformatic tools, and the strain’s phylogenetic relationships determined. Several specialized metabolite biosynthetic gene clusters were then identified and mutant strains were constructed to characterize NRPS cluster and their role in anti-complement agent’s production.

Although an extensive work has been carried out, for most part logically planned and carried out, several aspects are not thoroughly explained and prevent publication of this work in its present state

Questions and remarks:

- The manuscript suffers from poor English and requires careful proofreading. It contains errors as well as repeated ideas. Examples: line 63, “screening of microbial strains producing anti-complement compounds in our group revealed the potential of various marine-derived streptomycetes to produce anti-complement agents”

- The work is very much centred in anti-complement activities. As the authors say, it’s not very common to have streptomycetes producing these compounds so they should be better explained

- Line 79, 83, etc – ds is not the official abbreviation for days, replace by “days.

- Line 85 – the purpose of the E.coli culture explained here should be mentioned otherwise is no immediately clear why it was used (I’m guessing is the RecET protocol, gene deletion)

- Line 141, reference to intergeneric conjugation should be added

- The use of marine water fermentation should be further explored/explained apart from what is said in the abstract and conclusions

- How do the results obtained with the 16S rRNA analysis compare to the whole genome analysis? Does the same S. badius strain appear in the OAT heat map? What is the ANI value with this strain? What can be said about the use of 16S rRNA and whole genome for phylogenetic relationship determination?

- As BGC identification is part of the results and lead to cluster 21, this cluster should described in the results section. Then, if the authors think suitable, it can also be further developed in the discussion

- Line 288 – authors state that the ACA from the studied strain should be a polar compound. This should be better explained, that this conclusion was reached because only the methanol extract showed activity. This is then explained in the discussion but the first time its mentioned is not immediately clear.

- In figure 4 the negative controls of PCR with the wt and the same primers should be included.

- Why RiPPs were not explored as possible BGCs responsible for the anti-complement activity observed?

- There are two references Wang et al. 2016 that should be clearly distinguishable in text (for instance, 2016a and 2016b)

- Line 344 – what is the advantage of using this strain as a production host, specifically as an alternative to the already described and available Streptomyces strains for heterologous expression? Is the strain’s tolerance to higher salt concentrations a factor?

- Overall the discussion should be revised. It contains too many details and it should not be a repetition of the results.

Reviewer 2 ·

Basic reporting

The manuscript is well written, but, few concerns has to be pointed by this reviewer.
At line 187 ,the authors state “...based on the 16S rRNA gene sequences revealed that Streptomyces sp. S063 may be a subspecies of S. badius NRRL B-2567T (100% similarity)”
Inference of kinship level using only one marker is very bold. Unfortunately, only 16s does not gives you such resolution. Besides The figure 2 does present any support for the branches, nor, the methology for this inference are present on material and methods. In this case, the authors should use internal transcribed spacer (ITS) a very currently used genetic marker for infering relationship (Check PNAS April 17, 2012 109 (16) 6241-6246; https://doi.org/10.1073/pnas.1117018109).


On General genome information section, line 198 it is state that “...most abundant categories are transcription (683), 199 carbohydrate transport and metabolism (456) and amino acid transport and metabolism (462)”. This is indeed expected, since those are on lower levels (generic) ontologies. An gene ontology enrichment analyses, such as topGO (https://bioconductor.org/packages/release/bioc/html/topGO.html) might shed some lights on other high level (more specific) ontologies.

On lines 337 – 341 the authors selected four NRPS related BGCs but no result arise from this, so, why choose those? It was an empirical choice or there are some information available throughout the available literature to sustain this negative result? The last sentence starting on line 341 does not make any sense! The authors indicate that “Despite the negative results, we have demonstrated in this study that Streptomyces sp. S063 is easy to be genetically manipulated”. Compared to what organism? Wich data presented on the current manuscript lead to this finding?

On lines 349 – 351 the authors affirm “The rapid growth and abundant spores suggest Streptomyces sp. S063 is an optimal host for heterologous expression of BGCs”. As in previous comment this phrase is made in comparison to what? Other Streptomyces? Please, review your comparisons stated throughout the text, you always need to indicate to what are you comparing to.

On line 352 there is no clear cause/consequence on the following statement, regarding the growth on starch: “In addition, three cellulose-binding proteins and eight putative cellulases were also identified”. Is there some correlation present on the literature involving the presence of CBP and starch tolerance? This will enhance your discussion.

At lines 356 - 358 the phrases are also not much supported by the results presented on the current form of the manuscript. In this reviewer point of view this part might be suppressed from the text, or a revision must be done in a way to better conceptualize this affirmation.

Experimental design

The manuscript fits on the journal Scope. I felt that the discussion is quite superficial in some points, the authors indicate, more than once throughout the text, that some particular points need "further analyses". It leads the reader to a not a rigorous investigation feeling.
There is no methodology for the evolutionary 16s assessment present on figure 2.

Validity of the findings

The authors speculate more than their results are open to. Some comparisons seems more an affirmation than a comparison. And some of them are not supported by their results.

Additional comments

The current reviewed manuscript entitled “Analysis of the complete genome sequence of a marine streptomycete Streptomyces sp. S063 CGMCC 14582 reveals its biosynthetic potential to produce novel anti-complement agents and peptides” present the Streptomyces sp. S063 CGMCC 14582 genome sequencing and some annotation, also, present a hypothesis of ancestry between the under research specimen with Streptomyces badius NRRL B-2567. In this reviewer point of view, the authors should revisit their results and improve phylogenetics proposition adding some branch support and explain how the inference was made.

External reviews were received for this submission. These reviews were used by the Editor when they made their decision, and can be downloaded below.

---

## Round 0.2 · accepted · Accept

The authors have improved the manuscript following reviewers' suggestions.

Please just fix the typo in the title ("analysisof").

# Reviewer 1 ·

Basic reporting

no comment

Experimental design

no comment

Validity of the findings

no comment

Additional comments

This reviewer’s questions have been satisfactorily answered. The authors have made significant changes in the manuscript improving its overall quality. I have no further comments and support this work for publication in its present state.

Reviewer 2 ·

Basic reporting

The english has been improved. My criticisms regarding superficial discussion is still on. But all the major flaws has been covered after first round of revision. This manuscript is more suitable for a descriptive manuscript than able to present novel findings to the area, and, might by eligible for publication under this descriptive bias

Experimental design

The manuscript fits on the journal Scope. I still felt that the discussion is, somewhat, superficial in some points, even after previous criticisms the authors maintains that some particular points need "further analyses", even on the rebuttal letter. It leads the reader to a not a rigorous investigation feeling.

Validity of the findings

The overall presentation has been enhanced on this version, but the authors speculate more than their results are open to. It is OK for an descriptive paper.

External reviews were received for this submission. These reviews were used by the Editor when they made their decision, and can be downloaded below.